# Enhancing Public Participation in Public Health Offerings: Patient Preferences for Facilities in the Western Cape Province Using a Discrete Choice Experiment

**DOI:** 10.3390/ijerph19010590

**Published:** 2022-01-05

**Authors:** Plaxcedes Chiwire, Charlotte Beaudart, Silvia M. Evers, Hassan Mahomed, Mickaël Hiligsmann

**Affiliations:** 1Department of Health Services Research, CAPHRI Care and Public Health Research Institute, Maastricht University, 6200 MD Maastricht, The Netherlands; c.beaudart@maastrichtuniversity.nl (C.B.); s.evers@maastrichtuniversity.nl (S.M.E.); m.hiligsmann@maastrichtuniversity.nl (M.H.); 2Metro Health Services, Western Cape Government: Health and Division of Health Systems and Public Health, Department of Global Health, Faculty of Medicine and Health Sciences, Stellenbosch University, Stellenbosch 7505, South Africa; Hassan.Mahomed@westerncape.gov.za

**Keywords:** discrete choice experiments, patient preferences, facility choice, health provider choice, stated preferences

## Abstract

Understanding patients’ preferences for health facilities could help decision makers in designing patient-centered services. Therefore, this study aims to understand how patients’ willingness to trade for certain attributes affects the choice of public health facilities in the Western Cape province of Cape Town, South Africa. A discrete choice experiment was conducted in two community day centers (CDCs). Patients repetitively chose between two hypothetical health facilities that differed in six attributes: distance to facility, treatment by doctors vs. nurses, confidentiality during treatment, availability of medication, first visit (drop-in) waiting times, and appointment waiting times. The sample consisted of 463 participants. The findings showed that availability of medication (50.5%), appointment waiting times (19.5%), and first visit waiting times (10.2%) were the most important factors for patients when choosing a health facility. In addition, respondents preferred shorter appointment and first visit waiting times (<2 h). These results identified important characteristics in choosing public health facilities in Cape Town. These public health facilities could be improved by including patient voices to inform operational and policy decisions in a low-income setting.

## 1. Introduction

When people are unwell, the choice of where to seek medical care is influenced mainly by personal preferences, illness severity, and economic capacity [1,2]. The provision of quality health services is largely dependent on the sufficiency of the health workforce (in terms of numbers, the quality of skills they possess, how and where they are deployed, and how they are managed) [3]. The World Health Organization (WHO) argues that health systems are made of different components (organizations, institutions, and resources) that devote themselves to producing actions whose primary purpose is to improve health [4]. Most of the South African population access health services through government-run public clinics and hospitals. As South Africa moves to adopt National Health Insurance (NHI), the success of this process hinges on public healthcare reforms, which are critical to the delivery of high-quality, accessible, public-sector health services for universal coverage in the health system. Regardless of whether healthcare services become more affordable and available, people will not use them if the quality of these services is unacceptable [5]. Citizenry criticism of public healthcare is widely documented. As a result, several initiatives such as NHI Public Hearings have been carried out to capture concerns and try to improve the services offered by the government [6,7].

Public participation is widely regarded as the backbone of democracy in South Africa [8]. Several studies in the literature [3,8,9,10,11] have cited public participation as an instrument for establishing democracy and promoting unity between the government and the people. The aim of shifting decision making to a local level is to give citizens and their local representatives more power in public decision making. Thus, there is more participation in smaller geographic areas where mutual knowledge is greater, distances are shorter, and scales are smaller [11]. This could grant citizens greater influence on the formulation and implementation of health policies in the context of the national policy and healthcare framework, and the health system.

Community participation has widely been argued to be an important factor in improving health outcomes and the performance of health systems [10]. A relational skillset of intangible software, such as values, power, and communication, has been deemed crucial in fostering better community participation in health systems [11]. The inclusion of patient voices in policymaking is also increasingly gaining momentum in health technology assessments (HTA) [5,12,13,14,15]. Some studies note a lack of patient input in operations (day-to-day management) and treatment decisions, with healthcare providers continuing to make decisions for the provision of care on behalf of the patient [13,16,17]. It is essential to include patients, who are the users of health technology, in policymaking and day-to-day operational assessments. The outcomes following patient voice inclusion are likely to be more positive and beneficial for citizens. It is therefore important to elicit the preferences of patients using public health facilities. Several studies in the literature [3,5,8,10,18,19,20,21,22,23,24] argued for the need for more research to be carried out to integrate community perspectives into the available evidence-based health systems.

The National Core Standards for health establishments in South Africa specifies six priority areas for improvement: staff values and attitudes, waiting times, cleanliness, patient safety, security, infection prevention, control, and availability of medicines and supplies [25]. Information on how patients value these priority areas and other factors are often not presented in a format that shows their conditional relative importance when weighed against each other. For this reason, there is limited knowledge of South African patients’ preference weighting about choosing public health facilities. This is an important issue that influences decision making by management. The current paper, therefore, uses a discrete choice experiment to understand the conditional relative importance of various healthcare factors.

Discrete choice experiments (DCEs) are a stated preference method that has gained popularity when eliciting preferences for healthcare interventions and services. Even though there is a large body of work on stated preferences, only a few studies have been conducted regarding patients’ choice of facility, specifically within public health care. A systematic review of DCEs used to elicit patient preferences for primary healthcare showed that most studies were American or European, and focused on general practitioner (GP) consultations, [26] while Chinese studies appeared to focus primarily on the public health preferences of rural populations [27,28,29,30]. Only one South African DCE of public health facility preferences was identified [5]. This study suggested that treatment by doctors versus nurses, availability of medication, staff attitudes, waiting times, transport costs, expert advice, and examination are important factors influencing the choice of public health facilities. To the best of our knowledge, our study is the second DCE carried out in South Africa, focusing on patients attending public health facilities, contrary to Honda et al. (2015), who interviewed people outside the facility [5].

The present study aims to incorporate patient voices to identify areas in which the experience of care at public health facilities can be improved. The purpose is to facilitate demand for services and to increase patient satisfaction. The study uses a DCE to address patient preferences in the light of the proposed implementation of National Health Insurance in South Africa [31].

## 2. Methods

### 2.1. Discrete Choice Experiments

Research regarding patients’ preferences of the attributes mentioned above has been limited to quantitative and qualitative studies that look at the patients’ opinions of each attribute separately. Therefore, a DCE study was undertaken with the aim of better understanding patients’ willingness to trade off some of these attributes, which may affect their choice and attendance at public health facilities.

### 2.2. Study Design and Sampling

#### Selection of Attributes and Attribute Levels

The study followed the International Society for Pharmacoeconomics and Outcomes Research (ISPOR) conjoint analysis guidelines [32]. The guidelines provide researchers with the necessary steps for conducting a conjoint analysis. A step-wise approach [33] was followed when selecting attributes and attribute levels. This multi-step approach included a literature review, stakeholder consultations, and focus group discussions (FGDs) using the nominal group technique (NGT). The identification and prioritization of attributes related to choosing healthcare facilities are reported in Chiwire et al. (2021) [34]. The same identification and prioritization protocols were used for this DCE. Five candidate attributes were chosen as top priorities following the participants’ responses to the ranking and weighting process: distance ranked top of the list, followed by treatment by doctors, confidentiality during treatment, availability of medication, waiting times, and treatment by nurses, respectively. The attributes were divided into structural dimensions (distance to facility, waiting times, availability of drugs) and process dimensions (confidentiality during treatment and treatment by nurses or doctors) as per Donabedian’s healthcare quality model [26].

The list of attributes and attribute levels was finalized through consultation with the research team and facilities management team. For example, the teams agreed to split waiting times into two attributes due to the nature of the services offered at the Community Day Centers (CDCs)—due to high demand, patients can be instructed to return on a different day so that more urgent cases can be prioritized. Additionally, chronically ill patients presenting for their bi-annual check-up were most likely to have an appointment. Table 1 details the final list of attributes and levels.

### 2.3. Questionnaire Design and Sample Size

As it is not very efficient to provide all possible combinations of these patient attributes, a Bayesian D-efficient statistical design (Ngene software) was used to reduce the number of choice sets. In addition, the design aimed to maximize the precision of the estimated parameters for a given number of choice questions by including a priori information about the sign and value of the parameters.

In total, 24 binary choice sets were developed and divided into two versions, namely, questionnaire 1 with 12 choice sets and questionnaire 2 with the remaining 12. Participants were required to choose the alternative they preferred: facility A or facility B. Thus, each patient received 12 choice sets. Visual representations were used to facilitate patient understanding. The questionnaire was initially developed in English and then translated into Afrikaans and Xhosa. Figure 1 shows an example of a choice set.

The final questionnaire included the DCE, a willingness-to-pay section following another paper [35], and sociodemographic information. The DCE section started with a description of the task, a list of all attributes and levels, and an example of a choice task. A pilot study was conducted for face validity with 7 participants at Bothasig CDC to determine the feasibility of using the questionnaires in their original form, and to estimate the ease at which the participants could answer the questions. Only a few minor changes were made. The questionnaires were handed out to the clients.

Regarding sample size, the requirements of DCEs are not uniformly determined. The Lancsar and Louver method [36], advises at least 20–30 respondents/observations per choice set to provide precise parameter estimates. To strengthen the internal validity of the results, the literature recommends increasing sample size as well as the number of hypothetical scenarios [36,37,38]. Thus, guided by other studies, and to reduce sampling error, 200–250 respondents were decided to be sufficient for this study. Hence, a sample of 500 was recruited, with 250 participants at each facility.

### 2.4. Setting and Participants

The DCE was conducted in the South African city of Cape Town. The target population was clientele of primary health facilities at community day centers (CDCs) in the Northern and Tygerberg sub-structures (a part of the city containing 2 out of 8 subdistricts). The studied population and health facilities were identified through consultation with the Northern–Tygerberg substructure team, comprised of the sub-structure director and 3 primary health care managers. Following this, Goodwood CDC and Bothasig CDC were selected as study sites. Bothasig is in a more affluent area compared with Goodwood. The sites were also chosen due to similarities in their offered care package, reducing selection bias in facility choice. Community day centers (CDCs) in South Africa offer a comprehensive primary health care package (antenatal care, termination of pregnancy, reproductive health, chronic diseases and care, TB care, People with HIV/AIDS, mental health, oral health, rehabilitation and disability services, environmental health, occupational health, casualty, and maternity).

The participants were adult males and females aged 18 or above. All were patients accessing the Bothasig or Goodwood CDC public health facilities for any of the aforementioned services which are in the primary health comprehensive package. The participants were approached as they waited for consultation in the reception, pharmacy, or doctors’ waiting rooms. Some patients declined participation due to fear of missing their consultation; these participants were replaced. The sample is thus a convenience sample in response to the health services’ specific target. The findings would provide more broadly applicable information suited for decision making. It could also guide the selection of a more considerable, more representative study within the Western Cape.

## 3. Analysis

Analysis of the DCE was carried out using Nlogit software, version 5.0 (Econometric Software, Inc, NY, USA). First, a random parameter logit model or a latent class model were chosen for analyzing the choice observations. A random parameter logit model assumes that parameters are randomly distributed in the population, and captures heterogeneity by estimating the standard deviation of the parameters’ distribution [5,39,40,41]. All attributes were categorical. The constant was included in the model to test for a systematic preference for either facility. Second, all parameters were specified as random (i.e., normal distributions) to account for heterogeneity, and 2 000 Halton draws were conducted.

Effect coding was used to describe all categorical attributes. The reference/omitted levels were distance over 10 km; people can hear the conversation with nurses; given a date for appointment; waiting time with an appointment—8 h; treatment by a nurse; and some of the required medication is available. These were calculated as a negative sum of the attribute levels’ non-omitted coefficients and normalized to zero. The preference weights are relative to the mean effect of the different attribute levels and coefficients signs; the attribute preferences are either positive or negative compared with the mean. The conditional relative importance of each attribute was estimated using the beta-coefficients range for each attribute. Thereafter, interaction model analyses were conducted for sub-groups related to sociodemographic variables (age, gender, and facility).

## 4. Ethical Considerations

The Health Research Ethics Committee of Stellenbosch University approved the study. In line with the Western Cape Government’s research guidelines, approval for facilities access was granted by the Western Cape Government Provincial Health Research Committee. The study adhered to the Declaration of Helsinki principles. All participants were required to sign consent forms before completing the questionnaires.

## 5. Results

A total of 500 participants completed the DCE questions. Questionnaires that were incomplete, or filled out by participants who were under 18, were not included. This resulted in a final sample of 463 (232 at Bothasig and 231 at Goodwood). Unfortunately, no information about the response rate was recorded. Overall, the respondents’ demographic characteristics showed that most of the participants were female (61%), 35 years or above (59%), educated to grade 8–12 (high school) (64%), and unemployed (44%). Please refer to Table 2 for more information.

Participant characteristics were similar at both facilities. In both cases, more females than males answered the questionnaire. Most of the participants were married or in a partnership (46% at both facilities), and the proportion of singles was similar (35% and 36%, respectively). Students accounted for 7% at both facilities. Most participants traveled for 15–30 min to reach their local facility. However, several characteristics also differed between facilities: fifty percent of participants at Goodwood were 18–34 years old, while at Bothasig the majority was 35 or above. Half of the participants at Goodwood were unemployed, while at Bothasig, 64% were either formally employed or self-employed. Most participants at Bothasig were seeking care for a one-off condition, while at Goodwood, there were similar proportions of one-off patients and chronically ill patients. The visiting frequency was most commonly 1–12 months at Bothasig and >2 years at Goodwood. The mode of transport was most commonly walking for Goodwood participants, and private car for Bothasig participants. Please refer to Appendix A and Appendix A for more information.

## 6. Patient Preferences

The main results of the patients’ preferences, obtained using a random parameters logit model, are presented in Table 3 and Figure 2. At least one level of each attribute was significant at *p* < 0.05. An assessment of the attributes according to conditional relative importance shows that the availability of medication (50.5%) was the most important attribute for patients when selecting a facility, followed by appointment waiting times (19.5%), and first visit waiting times (10.4%). Conversely, going by relative importance, the least important attributes when selecting a facility were treatment by doctors vs. nurses (8.2%), distance to the health facility (6.7%), and confidentiality (4.6%).

Respondents preferred facilities where all required medication was available to facilities where only some medication was available. Respondents also preferred short appointment waiting times (2 h), with preference reducing as the waiting time increased. Similar findings were also observed for first visits, where short waiting times (2 h) were preferred to longer ones. In addition, respondents preferred to be treated by doctors rather than nurses. They also preferred absolute confidentiality—no one else being able to hear their consultation—compared with no confidentiality. Finally, the positive coefficient when the attribute of distance to the health facility was less than 3 km shows that respondents preferred traveling shorter distances, rather than longer distances. Standard deviations were, however, significant for all attributes except for treatment by doctors vs. nurses, showing that there was significant variation/heterogeneity within each attribute/level across respondents.

## 7. Sub-Group Analysis

Sub-group analysis was conducted on age, gender, and facilities. Significant differences were observed for the gender and facilities classes but not for the age groups.

### 7.1. Males vs. Females

The health facility preferences by males and females are presented in Table 4 and Appendix C. The interaction model revealed significant differences in waiting times and confidentiality. Females had slightly more preference for first visits and appointment shorter waiting times (2 h) than men. In addition, females reported a higher preference for confidentiality compared with men.

### 7.2. Facilities: Goodwood vs. Bothasig

Table 5 and Appendix D show the results for the facilities sub-group analysis. Goodwood participants had a much higher relative importance for medication availability (62%) than Bothasig (19%). Other attributes’ relative importance was internally evenly spread for Bothasig except for confidentiality. The interaction model revealed significant differences in the availability of medication. Goodwood had a significantly stronger preference for all the medication being available (*p* = 0.00) compared with Bothasig. Similarly, for most of the medication being available (*p* = 0.03). Bothasig had a significantly higher preference for treatment to be offered by a doctor (*p* = 0.02) than Goodwood.

The distance between 3–5 km was significantly less preferred at Goodwood compared with Bothasig. Respondents at Goodwood showed significantly stronger preference for confidentiality than at Bothasig. Both facilities preferred short first visit waiting times (2 h), with Bothasig having slightly more preference. As the times increased to 6 h, both facilities reduced preference, with a stronger negative preference at Goodwood than Bothasig.

## 8. Discussion

The present study aimed to incorporate patient voices to identify areas in which the experience of care at public health facilities can be improved. The purpose is to facilitate demand for services, and to increase patient satisfaction. The study uses a DCE to address patient preferences in the light of the proposed implementation of National Health Insurance in South Africa [31]. We managed to identify the most preferred attributes in choosing a facility, bridging the gap in the stated preferences on the topic studied in South Africa. Ours is the second DCE study investigating understanding public facility choice in the Western Cape and South Africa. In addition, there was strength in including a sample of patients at the facilities as participants who captured preferences of individuals experiencing and were able to attest to the service provided at the facility, unlike the Honda et al. [5] study that facilitated a DCE with the community setting.

The availability of medication was the most important relative attribute when selecting a health facility. The findings were consistent with Honda et al. [5], of which drug availability was identified as the most important issue. The literature demonstrates treatment measures to be the most important factors that affect healthcare seeking [28]. This study reveals that patients have preferences for certain characteristics of health facilities: near (short distances to health facility), with absolute confidentiality during visits, with short first visit waiting and appointment waiting time, where treatment is offered by doctors, and where all required medication is available.

Distance to health facilities has been argued to influence major health outcomes [42,43,44,45,46]. Universal access to health care requires service availability and accessibility. Therefore, distance to health facilities is a critical component of accessibility. Our study findings showed that patients generally prefer health facilities near households across facilities and demographic characteristics. However, studies in literature [42,44,45,46] showed that the relationship between distance and facility selection in urban settings could be less clear as women were cited as having more health service options within reasonable travel distance compared with men. The latter can explain higher preferences for facilities that are close by.

Our study findings also revealed that participants preferred to be treated by a doctor than a nurse; similar results were also recorded in a DCE by Caldow et al. [47], who reported that it is most important for respondents to see a general practitioner (GP) rather than a practice nurse. In our study, women reported appointment waiting time as the most important attribute when selecting a health facility. A previous DCE supported this, with a cohort of women that noted appraisals of the quality of care depended vigorously on the care process and nature of the services received rather than infrastructure [48].

Pedersen et al. [49], in their DCE, identified that patients preferred short waiting times if one had an appointment. This was also true in our study for both first visit waiting time and appointment waiting time. In addition, this attribute was cited as more important than “distance to the practice” in a patient preference study [49]. This was also true in our study as both first visit waiting time and appointment waiting time had higher relative importance proportions than the distance to health facility attribute. In terms of waiting time, all other things being equal, patients are generally less likely to choose healthcare services with long waiting times. The finding that women more than men prefer shorter waiting times appears to be unique to our study. Literature search on similarly findings showed gender gap on waiting times is mostly concentrated on surgical waiting times. We therefore did not have any comparable study. However, we can assume this may be linked to the amount of responsibilities carried by women in households. 

The structural outcomes at Goodwood were of greater concern compared with the Bothasig. Goodwood seemed to lack resources (based on observation or the participants’ responses), affecting their structural outcomes. The interaction model for facilities revealed significant differences between Goodwood and Bothasig participants. More specifically, Goodwood participants gave more importance to the availability of medication, confidentiality, shorter distances, and first visit waiting times. Medication availability could suggest concern over stock-outs that require attention, re-assessing drug supply chain, and home delivery systems. There have been reported stock-outs in South Africa, mainly for HIV and TB drugs, associated with the scale-up of treatments [50]. As noted in Chiwire et al. [34], the focus group discussions process for selecting attributes for this study revealed an overcrowded Goodwood facility with less confidentiality during nurses’ first point of patient screening. It is not surprising that the same issues were considered most important at Goodwood. Bothasig facility was less crowded and appeared to provide more confidentiality during screening. It is recommended that infrastructure and patient flow at Goodwood be re-assessed. Alternative methods to reduce overcrowding and long waiting times apart from the current deferral appointment system should also be considered. Despite the national policy on managing patient waiting time in outpatient departments [51], improvements in waiting times for first and appointment visits appear to be slow. They need to be continuously monitored and strengthened. More participants at Bothasig were educated at the diploma level and above (34.3%) compared with 16.9% at Goodwood. There is a possibility of a correlation between education and the higher preference for treatment by a doctor at Bothasig.

This study has several limitations. The differences in facility layout and patient flow may have increased bias towards preferences from participants. Goodwood facility layout does not allow for confidentiality especially at first point of contact between health professions and patients. The space is not big enough to have the first point contact consultations to be conducted separately, hence nurses doing temperature screening and those doing medical probing will be in the same room. As a result, there were high patient volumes at Bothasig compared with Goodwood as a result of these structural differences. Secondly, external validity ensures comparability of hypothetical and actual choices [52]. As respondents are not obliged in reality to make the choices they indicate in a DCE, hypothetical bias may reduce the usefulness of DCE results [53]. However, the results in our study are not far removed from other findings and policy-targeted priority areas [25,51,54]. Lastly, facets of participants who refused to participate were not systematically collected and we were not able to do a contrast with those who participated.

The study brings in a wealth of knowledge especially in the Southern African context and specifically South Africa. To date, very few well-founded scientific studies have been conducted in South Africa and we are aware of only one study in the South African context that looked at patients’ preferences especially from a trade-off point of view in the public health facility, and it concentrated on a community sample. Therefore, this study provides relevant scientific valuable information to policymakers as South Africa, like any other low-middle income country, is characterized by a limited health budget. Our study is further innovative in the sense that we reveal that patients are willing to accept trade-offs between the included attributes and most were thus important. Considering alternative data sources available to decision makers is important, more so for them to understand how useful DCEs are in predicting behavior. The quantification of how well DCEs predict behavior could explicitly account for uncertainty in DCE predictions [53]. DCEs can provide a relatively accurate and cost-effective option to predict individual choices [53]. The data from DCEs can then be used to quantify the relative importance of aspects of health care [48]. Therefore, this study avails information to policymakers on patients’ preferences in the Western Cape, which is relatively accurate. Thus, accounting for the variation in DCE prediction accuracy in this manner would make for more robust uptake and impact models.

## 9. Conclusions

The study findings show overall availability of medication is the most important factor in choosing a facility for service provision. Shorter waiting times were preferred either on appointment or first visit. Being treated by doctors was significantly preferred to being treated by nurses, while the shorter distance to facility and confidentiality were highly preferred. Decision makers must include these patient voices in improving healthcare provision and increasing patient satisfaction.

## Figures and Tables

**Figure 1 ijerph-19-00590-f001:**
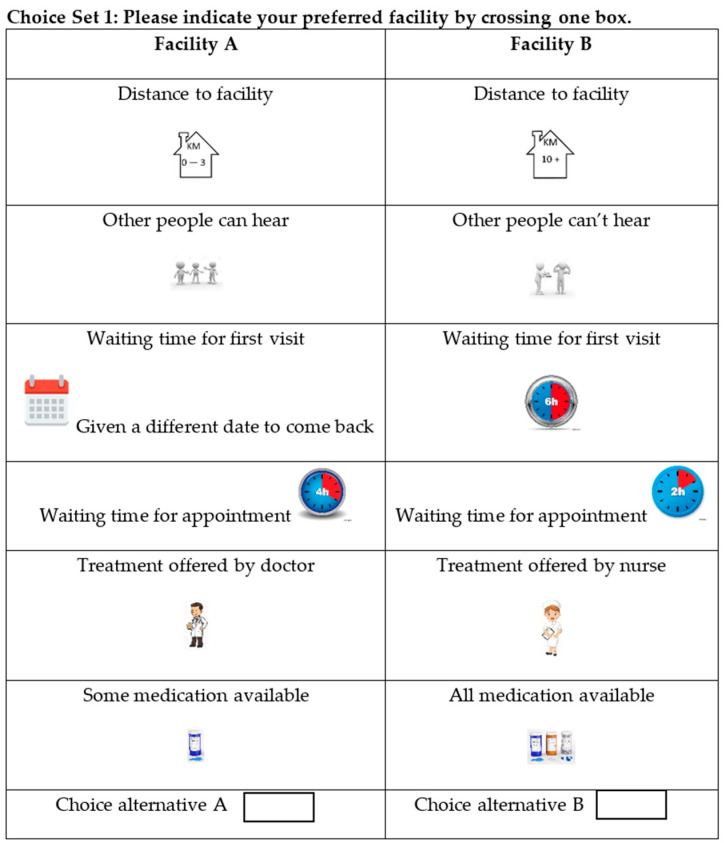
Example of a choice set in the DCE questionnaires.

**Figure 2 ijerph-19-00590-f002:**
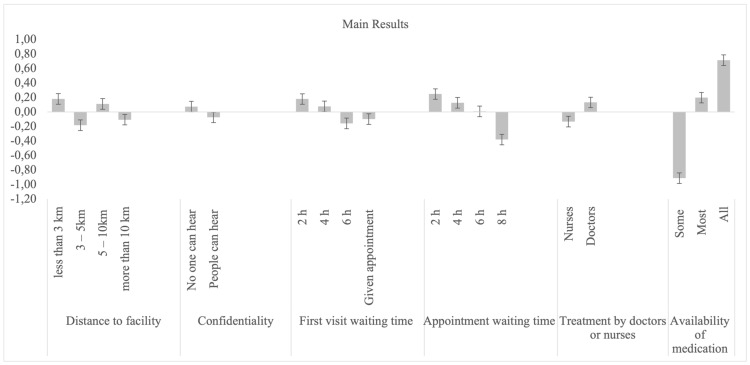
Main results random parameters logit model with a standard deviation.

**Table 1 ijerph-19-00590-t001:** Attributes, definition, and attribute levels describing facility choice preferences in the DCE.

Attributes	Definition of Attributes	Attribute
Distance to CDC	How far for patients to travel to the community day center (CDC) in kilometers from their home.	Less than 3 km
		Between 3–5 km
		Between 5–10 km
		More than 10 km
Confidentiality during treatment	Consultation is carried out behind closed doors, without other patients and staff hearing.	Other patients and staff cannot hear the consultation
		Other patients and staff can hear the consultation
Waiting time for the first visit to the facility (without an appointment)	How long does it take to consult with the doctor or nurse after entering the CDC. If the first visit is for a particular ailment, they will give a diagnosis.	2 h
		4 h
		6 h
		Different day than the appointment
Waiting time with an appointment	How long does it take to see the doctor or nurse after entering the CDC with a pre-booked appointment?	2 h
		4 h
		6 h
		8 h
Treatment offered by	Which staff member provides a diagnosis.	Doctor
		Nurse
Availability of required medication	The patient receives the medication prescribed at the CDC.	All the required medication is available
		Most of the required medication is available
		Some of the required medication is available

**Table 2 ijerph-19-00590-t002:** Respondents’ demographic characteristics.

Type of Factors	Variables	All (n = 466)	Facility		
			Bothasig CDC (n = 230)	Goodwood CDC (n = 236)	*p*-Value
**Sociodemographic Factors**	**Sex (%)**				
	Male	179 (38.4)	81 (35.2)	98 (41.5)	0.024
	Female	282 (60.5)	149 (64.8)	133 (56.4)	
	**Age Group (%)**				
	18–34	192 (41.2)	114 (49.6)	78 (33.1)	<0.001
	35+	274 (58.8)	116 (50.4)	158 (66.9)	
	**Marital Status (%)**				
	Single	164 (35.2)	82 (35.7)	82 (34.7)	0.931
	Married or partnership	214 (45.9)	106 (46.1)	108 (45.8)	
	Widowed	28 (6.0)	13 (5.7)	15 (6.4)	
	Divorced	46 (9.9)	22 (9.6)	24 (10.2)	
	Separated	6 (1.3)	4 (1.7)	2 (0.8)	
	**Education (%)**				
	Grade 0–7	32 (6.9)	12 (5.2)	20 (8.5)	0.022
	Grade 8–12	296 (63.5)	137 (59.6)	159 (67.4)	
	National Qualifications Framework (NQF, diploma or certificate)	80 (17.2)	52 (22.6)	28 (11.9)	
	Bachelor’s degree	43 (9.2)	24 (10.4)	19 (8.1)	
	Master’s degree	8 (1.7)	3 (1.3)	5 (2.1)	
	**Employment (%)**				
	Student	34 (7.3)	17 (7.4)	17 (7.2)	0.034
	Unemployed	205 (44.0)	86 (37.4)	119 (50.4)	
	Self-employed	53 (11.4)	31 (13.5)	22 (9.3)	
	Employed	165 (35.4)	93 (40.4)	72 (30.5)	
**Access-Related Factors**	**Facility Visit**				
	First time	54 (11.6)	37 (16.1)	17 (7.2)	<0.001
	Between 1–12 months	120 (25.8)	77 (33.5)	43 (18.2)	
	Between 1–2 years	74 (15.9)	48 (20.9)	26 (11.0)	
	2 years or more	208 (44.6)	65 (28.3)	143 (60.6)	
	**Chronic**				
	Chronic	203 (43.6)	89 (38.7)	114 (48.3)	0.097
	One-off	236 (50.6)	127 (55.2)	109 (46.2)	
	Both	10 (2.1)	4 (1.7)	6 (2.5)	
	**Transport**				
	Walking	186 (39.9)	58 (25.2)	128 (54.2)	<0.001
	Public taxi	72 (15.5)	31 (13.5)	41 (17.4)	
	Private taxi	12 (2.6)	7 (3.0)	5 (2.1)	
	Bus	16 (3.4)	12 (5.2)	4 (1.7)	
	Private car	170 (36.5)	119 (51.7)	51 (21.6)	
	Ambulance	1 (0.2)	1 (0.4)	0 (0.00)	
	**Travel Time**				
	0–15 minutes	246 (52.8)	128 (55.7)	118 (50.0)	0.208
	15–30 minutes	140 (30.0)	66 (28.7)	74 (31.4)	
	30 minutes to 1 h	51 (10.9)	19 (8.3)	32 (13.6)	
	More than 1 h	11 (2.4)	7 (3.0)	4 (1.7)	

*p*-values obtained with Chi^2^ test; 5 missing data for gender; 8 missing data for marital status; 7 missing data for education; 9 missing data for employment; 10 missing data for facility visit; 16 missing data for chronic; 9 missing data for transport; and 17 missing data for travel time (+1 data entry error/coded as 5).

**Table 3 ijerph-19-00590-t003:** Main results random parameters logit model.

Overall Results	Coefficient	Estimated Standard Deviation	Relative Importance
Attribute/Level			
Constant (non-random parameter)	0.07 ** (0.01 to 0.14)		
**Distance**			
Distance to facility is less than 3 km	0.18 *** (0.67 to 0.29)	0.23 ***	
Distance to facility is 3–5 km away from home	−0.18 *** (−0.27 to −0.10)	0.36 ***	
Distance to facility is 5–10 km away from home	0.11 *** (0.03 to 0.18)	0.19 **	
Distance to facility is more than 10 km	−0.11 ** (−0.21 to −0.01)		6.7%
**Confidentiality**			
No one can hear your conversation with the nurses	0.07 *** (0.02 to 0.13)	0.22 ***	
People can hear conversation with nurses	−0.07 (−0.13 to −0.02)		4.6%
**First visit waiting time**			
First visit waiting time 2 h	0.12 *** (0.08 to 0.27)	0.20 **	
First visit waiting time 4 h	0.08 * (−0.00 to 0.16)	0.30 ***	
First visit waiting time 6 h	−0.16 (−0.23 to −0.08)	0.1	
Given different date appointment	−0.10 (−0.22 to 0.02)		10.4%
**Appointment waiting time**			
Appointment waiting time 2 h	0.25 *** (0.14 to 0.35)	0.23 ***	
Appointment waiting time 4 h	0.13 *** (0.05 to 0.20)	0.19 **	
Appointment waiting time 6 h	0.01 (−0.07 to 0.09)	0.01	
Appointment waiting time 8 h	−0.38 *** (−0.49 to −0.27)		19.5%
**Treatment by doctors or nurses**			
Treatment offered by nurses	−0.13 (−0.23 to −0.04)		
Treatment offered by doctor	0.13*** (0.04 to 0.23)	0.1	8.2%
**Availability of medication**			
Some of the medication required is available	−0.91 *** (−1.10 to −0.72)		
Most of the required medication is available	0.20 (0.14 to 0.26)		
All medication required is available	0.71 *** (0.55 to 0.88)	0.97 ***	50.5%
Replications for simulated probs. = 1000	Log likelihood function −3618.69568	McFadden pseudo R-squared 0.0528530	
RPL model with panel has 466 groups	Restricted log likelihood −3820.62726	Estimation based on *n* = 5512, K = 27	
Fixed number of observations./group = 12	Chi squared [27] (*p* = 0.000) 403.86315	Inf.Cr.AIC = 7291.4 AIC/*n* = 1.323	
Number of observations.= 5592, skipped 80 observations	Significance level 0.00000		

*** Significant at 0.01; ** Significant at 0.05; and * Significant at 0.1.

**Table 4 ijerph-19-00590-t004:** Differences between male and female respondents in preferences for facilities in the Western Cape province.

Sub-Group Gender	Male			Female			Interaction Gender
Attribute/Level	Coefficient	Estimated Standard Deviation	Relative Importance	Coefficient	Estimated Standard Deviation	Relative Importance	Significance *p*-Value
Constant (non-random parameter)	0.07 (−0.03 to 0.18)			0.07 * (−0.01 to 0.16)			0.15
**Distance**							
Distance to facility is less than 3 km	0.09 (−0.10 to 0.28)	0.41 ***		0.24 *** (0.10 to 0.39)	0.15		0.18
Distance to facility is 3–5 km away from home	−0.16 ** (−0.29 to −0.02)	0.35 ***		−0.20 *** (−0.30 to −0.9)	0.36 ***		0.67
Distance to facility is 5–10 km away from home	0.13 ** (0.00 to 0.25)	0.17		0.09 (−0.00 to 0.19)	0.22 **		0.65
Distance to facility is more than 10 km	−0.06 (−0.22 to 0.10)		11.4%	−0.13 ** (−0.25 to −0.01)		8%	
**Confidentiality**							
No one can hear your conversation with the nurses	0.01 (−0.07 to 0.09)	0.18 ***		0.12 *** (0.05 to 0.19)	0.26 ***		0.06 *
People can hear conversation with nurses	−0.01 (−0.09 to 0.07)		0.8%	−0.12 (−0.19 to −0.05)		6%	
**First visit waiting time**							
First visit waiting time 2 h	0.05(−0.10 to 0.21)	0.24 **		0.27 *** (0.14 to 0.39)	0.20 *		0.03 **
First visit waiting time 4 h	0.17 ** (0.03 to 0.30)	0.34 ***		0.04 (−0.06 to 0.14)	0.25 ***		0.15
First visit waiting time 6 h	0.10 (−0.22 to 0.02)	0.11		0.19 *** (−0.09 to 0.11)	0.11		0.23
Given different date appointment	−0.32 ** (−0.53 to −0.11)		14.6%	−0.12 (−0.28 to 0.04)		12%	
**Appointment Waiting time**							
Appointment waiting time 2 h	0.10 (−0.63 to 0.27)	0.21		0.34 *** (0.20 to 0.47)	0.21		0.04 **
Appointment waiting time 4 h	0.14 ** (0.02 to 0.27)	0.34 ***		0.12 ** (0.02 to 0.22)	0.19		0.78
Appointment waiting time 6 h	−0.01 (0.14 to 0.12)	0.13		0.01 (−0.09 to 0.11)	0.00		0.88
Appointment waiting time 8 h	−0.23 ** (−0.41 to −0.05)		14.6%	−0.47 *** (−0.61 to −0.33)		21%	
**Treatment by doctors or nurses**							
Treatment offered by nurses	−0.07 (−0.23 to 0.08)			−0.17 (−0.30 to −0.05)			
Treatment offered by doctor	0.07 (−0.08 to 0.23)	0.13	5.5%	0.17 *** (0.05 to 0.30)	0.12	9%	0.34
**Availability of medication**							
Some of the medication required is available	−0.77 (0.47 to 1.07)			−0.87 (0.62 to 1.12)			
Most of the required medication is available	0.19 *** (0.10 to 0.29)	0.23 ***		0.10 *** (0.12 to 0.28)	0.29 ***		0.97
All medication required is available	0.58 *** (0.32 to 0.84)	0.98 ***	53.1%	0.77 *** (0.12 to 0.28)	1.00 ***	43%	0.28

*** Significant at 0.01; ** Significant at 0.05; and * Significant at 0.1.

**Table 5 ijerph-19-00590-t005:** Differences between Goodwood and Bothasig respondents in preferences for facilities in the Western Cape province.

Sub-Group Facilities	Goodwood			Bothasig			Interaction Facility
Attribute/Level	Coefficient	Estimated Standard Deviation	Relative Importance	Coefficient	Estimated Standard Deviation	Relative Importance	Significance *p*-Value
Constant (non-random parameter)	0.10 * (−0.00 to 0.19)			0.06 (−0.03 to 0.15)			0.17
**Distance**							
Distance to facility is less than 3 km	0.25 *** (0.82 to 0.42)	0.17		0.09 (−0.07 to 0.25)	0.32 ***		0.18
Distance to facility is 3–5 km away from home	−0.29 *** (−0.43 to −0.53)	0.63 ***		−0.07 (−0.18 to 0.04)	0.01		0.01 ***
Distance to facility is 5–10 km away from home	0.12 ** (0.01 to 0.23)	0.22 **		0.11 ** (0.00 to 0.22)	0.21 **		0.98
Distance to facility is more than 10 km	−0.08 (−0.23 to 0.07)		11.3%	−0.13 * (−0.27 to 0.01)		10%	
**Confidentiality**							
No one can hear your conversation with the nurses	0.16 *** (0.09 to 0.24)	0.08		0.00 (−0.08 to 0.08)	0.29 ***		0.02 **
People can hear conversation with nurses	−0.16 (−0.24 to −0.09)		6.7%	0.00 (−0.08 to 0.08)		-	
**First visit waiting time**							
First visit waiting time 2 h	0.07 (−0.06 to 0.21)	0.05		0.29 *** (0.15 to 0.44)	0.35 ***		0.02 **
First visit waiting time 4 h	0.11 * (−0.01 to 0.22)	0.17		0.03 (−0.09 to 0.15)	0.35 ***		0.21
First visit waiting time 6 h	−0.26 *** (−0.37 to −0.14)	0.04		−0.08 (−0.18 to 0.02)	0.14		0.03 **
Given different date appointment	0.08 (−0.10 to 0.26)		7.8%	−0.24 ** (−0.42 to −0.06)		22%	
**Appointment Waiting time**							
Appointment waiting time 2 h	0.20 *** (0.05 to 0.36)	0.33 ***	11.8%	0.28 *** (0.14 to 0.43)	0.19		0.30
Appointment waiting time 4 h	0.21 *** (0.09 to 0.33)	0.35 ***		0.08 (−0.03 to 0.19)	0.05		0.34
Appointment waiting time 6 h	−0.05 (−0.16 to 0.07)	0.02		0.07 (−0.04 to 0.18)	0.01		0.25
Appointment waiting time 8 h	−0.36 *** (−0.53 to −0.19)			−0.43 *** (−0.58 to −0.28)		29%	
**Treatment by doctors or nurses**							
Treatment offered by nurses	−0.01 (−0.16 to 0.13)			−0.24 (−0.38 to −0.10)			
Treatment offered by doctor	0.01 (−0.13 to 0.16)	0.00	0.4%	0.24 *** (0.10 to 0.38)	0.25 ***	20%	0.02 **
**Availability of medication**							
Some of the medication required is available	−1.62 (1.31 to 1.93)			−0.29 ** (0.06 to 0.52)			
Most of the required medication is available	0.29 *** (0.20 to 0.39)	0.34 ***		0.13 *** (0.04 to 0.21)	0.23 ***		0.03 **
All medication required is available	1.33 *** (1.06 to 1.61)	1.14 ***	62.0%	0.16 (−0.04 to 0.36)	0.65 ***	19%	0.00 ***

*** Significant at 0.01; ** Significant at 0.05; and * Significant at 0.1.

## Data Availability

The data presented in this study are available on request from the corresponding author.

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
