# Peer review of "Enhancing Public Participation in Public Health Offerings: Patient Preferences for Facilities in the Western Cape Province Using a Discrete Choice Experiment"

_ijerph, 2022, doi:10.3390/ijerph19010590_

Round 1

Reviewer 1 Report

This is an interesting study, but I'm not certain that the findings tell us anything that is not obvious and already known. Studies show that patients prefer healthcare providers that have their medications, lower waiting times, are closer to where they live, and so on. The authors should make it more obvious what their contributions to this field of study are. Otherwise, this study adds nothing new or helpful. 

There are some grammatical mistakes/concerns in this paper. For instance, in the 8th line of the Setting and Participants section, the words "compared to." 

  1. This paper has many grammatical problems. I gave just some examples below.

For instance, in the Introduction these sentences are clumsy and wordy: 

“In a nutshell, regardless of whether healthcare services become more affordable and available, unfortunately, people will not use them if the quality of these services is unacceptable. Citizenry criticism of public healthcare is widely documented, and is an issue that has required public participation hearings an example being NHI Public Hearings , as well as surveys of other methodologies to capture concerns and try to improve the services offered by the government.”

For instance, in the Results section, this sentence does not make sense:

“Finally, the positive coefficient when the attribute of distance to the health facility was less than 3 km shows that respondents preferred traveling shorter distances, rather than longer standard deviations were significant for all attributes except treatment by doctors vs. nurses, showing that there was significant variation within each attribute across respondents in the significant attributes.”

For instance, in the Discussion these sentences are not correct or are unclear:

“Women in our study as in both study settings are urban with access to facility being relatively easier than in rural areas.”

“The interaction model for facilities revealed Goodwood participants to have stronger preferences for the availability of medication, shorter appointment and first visit waiting times, confidentiality.”

“Other methods to reduce overcrowding time other than the current deferral appointment system should also be considered.”

  1. This study’s finding that women, more than men, prefer shorter waiting times. It would be interesting for the authors to try to explain this gender difference.

  1. The authors mentioned that the differences in facility layout and patient flow may have increased bias towards preferences from participants. I consider this to be a major limitation that needs to be addressed in more detail than it is in the current paper.

Author Response

We are grateful to both reviewers for dedicating their time and effort to review our manuscript and for the detailed feedback which has been availed to us.

Response to Reviewer 1 Comments

Point 1: This is an interesting study, but I'm not certain that the findings tell us anything that is not obvious and already known. Studies show that patients prefer healthcare providers that have their medications, lower waiting times, are closer to where they live, and so on. The authors should make it more obvious what their contributions to this field of study are. Otherwise, this study adds nothing new or helpful. 

Response 1: We acknowledge the argument but in our view the study brings in a wealth of knowledge especially in the Southern African context and specifically South Africa. To date, very few well-founded scientific studies have been done in South Africa and we are just aware of only one study in the South Africa context have looked at patients’ preferences especially from a trade-off point of view in the public health facility and it concentrated on a community sample. We think that our study provides relevant scientific information for policy makers, using patients attending the facilities. Our study is further innovative in the sense that we reveal that patients are willing to trade-offs between the included attributes and most were thus important. Further, the availability of medications emerge as the most important consideration for patients and a key message for policy makers, as all clinics does not have medications at the moment.

Point 2: This paper has many grammatical problems.

Response 2: We have since taken the manuscript for grammar editing with an English academic professional.All the grammar issues raised have been addressed and tracked changes.

I gave just some examples below.

For instance, in the Introduction these sentences are clumsy and wordy:

“In a nutshell, regardless of whether healthcare services become more affordable and available, unfortunately, people will not use them if the quality of these services is unacceptable. Citizenry criticism of public healthcare is widely documented and is an issue that has required public participation hearings an example being NHI Public Hearings, as well as surveys of other methodologies to capture concerns and try to improve the services offered by the government.”

Response: We have re-written the sentence as follows: Regardless of whether healthcare services become more affordable and available people, will not use them if the quality of these services is unacceptable [5]. Citizenry criticism of public healthcare is widely documented. As result, several initiatives such as NHI Public Hearings have been done to capture concerns and try to improve the services offered by the government [6,7]. 

For instance, in the Results section, this sentence does not make sense:

“Finally, the positive coefficient when the attribute of distance to the health facility was less than 3 km shows that respondents preferred traveling shorter distances, rather than longer standard deviations were significant for all attributes except treatment by doctors vs. nurses, showing that there was significant variation within each attribute across respondents in the significant attributes.”

Response: We have re-written the sentence as follows; “Finally, the positive coefficient when the attribute of distance to the health facility was less than 3 km shows that respondents preferred traveling shorter distances, rather than longer distance. Standard deviations were however significant for all attributes except for treatment by doctors vs. nurses, showing that there was significant variation/heterogeneity within each attribute/level across respondents.

For instance, in the Discussion these sentences are not correct or are unclear:

“Women in our study as in both study settings are urban with access to facility being relatively easier than in rural areas.”

Response: We have deleted this sentence. The sentence given before it sufficiently explains the point we wanted to put across.  

“The interaction model for facilities revealed Goodwood participants to have stronger preferences for the availability of medication, shorter appointment and first visit waiting times, confidentiality.”

Response: We have re-written the sentence as follows; “The interaction model for facilities revealed significant differences between Goodwood and Bothasig participants. More specifically, Goodwood participants gave more importance to the availability of medication, confidentiality, shorter distances and first visit waiting times.”

“Other methods to reduce overcrowding time other than the current deferral appointment system should also be considered.”

Response: We have re-written the sentence as follows; “Alternative methods to reduce overcrowding and long waiting times apart from the current deferral appointment system should also be considered”

Point 3: This study’s finding that women, more than men, prefer shorter waiting times. It would be interesting for the authors to try to explain this gender difference.

Response 3: We have since taken the suggestion into account and updated; “The finding that women more than men prefer shorter waiting times appears to be unique to our study. Literature search on similarly findings showed gender gap on waiting times is mostly concentrated on surgical waiting times. We therefore did not have any comparable study. However, we can assume this may be linked to the amount of responsibilities carried by women in households.” 

Point 4: The authors mentioned that the differences in facility layout and patient flow may have increased bias towards preferences from participants. I consider this to be a major limitation that needs to be addressed in more detail than it is in the current paper.

Response 4: We have since taken the suggestion into account and updated the limitation to reflect in detail how the differences in structural layout might pose bias. This is also why we conducted a subgroup analyses according to facilities. This is written as follows; “The differences in facility layout and patient flow may have increased bias towards preferences from participants. Goodwood facility layout does not allow for confidentiality especially at first point of contact between health professions and patients. The space is not big enough to have the first point contact consultations to be done separately, hence nurses doing temperature screening and those doing medical probing will be in the same room. As a result, they were high patient volumes at Bothasig compared to Goodwood as result of these structural differences.”

Reviewer 2 Report

Dear Authors

Congratulations for your good research and report. Your introduction gives me clear idea why you need to do that study. However, please allow me to share some area for improvement of your presentation in methods session.

 In your report about research method, under the sub heading Setting and Participants, I found  the following phase  "The findings would provide more broadly applicable information suited for decision-making. It could also guide the selection of a more considerable, more representative study within the Western Cape." and I felt that phrase is not related to your method of sampling or participant selection. Your sampling is convenient sampling and you mentioned that some patient declined to participate. If you can mention about response rate it will be beneficial for the reader.

In result session, you mentioned the result of subgroup analysis. I wonder is there any difference in preference of male and female patients regarding consultation with doctor or nurses. According to your table 4 there are differences if I am not mistaken but I could not find about it in your writing. I felt that fact is important as female dominates in nursing practices and most of the primary care clinic users are pregnant women and mothers. 

in your another sub group analysis you have mentioned that "Bothasig had a significantly higher preference for treatment to be offered by a doctor  (p = 0.02) than Goodwood". I wonder why and will you mention it in discussion? If you mention more about services offered in two CDC  ( eg. Antenatal care, post natal care, family planning, elderly care, out patient care) and whether your participants are recruited from all users of different services in both clinics the reader have some idea for the differences in consultation preferences among participants of Bothasig and Goodwood.

Author Response

Response to Reviewer 2 Comments

We are grateful to both reviewers for dedicating their time and effort to review our manuscript and for the detailed feedback which has been availed to us.

Point 1: Congratulations for your good research and report. Your introduction gives me clear idea why you need to do that study. However, please allow me to share some area for improvement of your presentation in methods session.

Response 1: We are grateful for the opportunity and time you have taken in reviewing our manuscript and value the great insight you have brought to the table.

Point 2: In your report about research method, under the sub heading Setting and Participants, I found the following phase “The findings would provide more broadly applicable information suited for decision-making. It could also guide the selection of a more considerable, more representative study within the Western Cape." and I felt that phrase is not related to your method of sampling or participant selection. Your sampling is convenient sampling and you mentioned that some patient declined to participate. If you can mention about response rate it will be beneficial for the reader.

Response 2: We have since updated the methods section to reflect the reviewer suggestions. Unfortunately, no information about the response rate was recorded, hence providing a response would only be speculative. However, we have added the following to the limitations: “Lastly, facets of participants who refused to participate were not systematically collected and we were not able to do a contrast with those who participated.”.

Point 3: In result session, you mentioned the result of subgroup analysis. I wonder is there any difference in preference of male and female patients regarding consultation with doctor or nurses. According to your table 4 there are differences if I am not mistaken but I could not find about it in your writing. I felt that fact is important as female dominates in nursing practices and most of the primary care clinic users are pregnant women and mothers.

Response 3: There was no significant difference in preference of male and female regarding consultation with doctor or nurses (p=0.35).  We therefore did not see the need to comment and report on them.

Point 4: in your another sub group analysis you have mentioned that "Bothasig had a significantly higher preference for treatment to be offered by a doctor (p = 0.02) than Goodwood". I wonder why and will you mention it in discussion? If you mention more about services offered in two CDC ( eg. Antenatal care, post natal care, family planning, elderly care, out patient care) and whether your participants are recruited from all users of different services in both clinics the reader have some idea for the differences in consultation preferences among participants of Bothasig and Goodwood.

Response 4: We have updated the methods to reflect concerns raised:

The link between education level and preference for doctors as follows in the discussion section: “More participants at Bothasig were educated at Diploma and above (34.3%) compared to 16.9% at Goodwood. There is a possibility there could be a correlation between education and the higher preference for treatment by a doctor at Bothasig.”

We have also added in more detail the services which constitute the comprehensive primary health care package. This was written as follows: “ Community Day Centres (CDCs) in South Africa, offer a comprehensive primary health care package (antenatal care, termination of pregnancy, reproductive health, chronic diseases and care, TB care, People with HIV/AIDS, mental health, oral health, rehabilitation and disability services, environmental health, occupational health, casualty, and maternity). The participants were adult males and females aged 18 or above. All were patients accessing the Bothasig or Goodwood CDC public health facilities for any of the aforementioned services which are in the primary health comprehensive package.”
